# Reducing dementia-related stigma and discrimination among community health workers in Brazil: protocol for a randomised controlled feasibility trial

Déborah Oliveira ![ORCID],[1] Carolina Godoy,[1] Fabiana A F da Mata,[1] Elaine Mateus,[2,3] Ana Carolina Arruda Franzon,[2] Nicolas Farina ![ORCID],[4] Sara Evans-Lacko,[5] Cleusa P Ferri[1,6]

For numbered affiliations see end of article.

**Correspondence to**
Dr Déborah Oliveira;
oliveiradc.phd@gmail.com

## ABSTRACT

**Introduction** Stigma and discrimination among healthcare workers can hinder diagnosis and the provision of appropriate care in dementia. This study is aimed at developing, delivering and evaluating the feasibility of a group antistigma intervention to improve knowledge, attitudes and behaviours in relation to people living with dementia among community health workers (CHWs).

**Methods and analysis** This will be a randomised controlled feasibility trial conducted with 150 CHWs from 14 primary care units (PCUs) in São Paulo, Brazil. PCUs will be randomly allocated (1:1) in two parallel groups—experimental group or control group. Participants from PCUs allocated to the experimental group will receive a 3-day group intervention involving audio-visual and printed materials as well as elements of social contact. The control group will keep their usual routine. Knowledge, attitude and intended behaviour stigma-based outcomes will be assessed at baseline and at follow-up (30 days after intervention) to both groups, with additional questions on feasibility for the experimental group at follow-up. Around 10–15 participants will take part in follow-up semistructured interviews to further explore feasibility. Quantitative analyses will follow an 'intention to treat' approach. Qualitative data will be analysed using content analysis.

**Ethics and dissemination** This study was approved by the National Commission for Ethics in Research in Brazil (n. 5.510.113). Every participant will sign a consent form. Results will be disseminated through academic journals and events related to dementia. The intervention materials will be made available online.

## STRENGTHS AND LIMITATIONS OF THIS STUDY

⇒ This is the first study to develop, deliver and evaluate a controlled feasibility study of a group antistigma intervention to improve knowledge, attitudes and behaviours in relation to people living with dementia among community health workers in Brazil.

⇒ This study will seek to involve all the community health workers of all the primary care units of a large urban city in São Paulo, Brazil.

⇒ The study includes quantitative and qualitative components that will assess the feasibility of the intervention from different perspectives and will include a follow-up assessment, which will help us understand any potential long-term impact.

⇒ As this will be a feasibility study, no conclusions can be drawn about the effectiveness of the intervention; however, we hope that the information collected will help build a robust future randomised controlled trial.

impact on the lives of those living with the condition and their families, such as by hindering access to diagnosis and support.[2 3] Combating the stigma related to dementia is a global health priority.[1]

Stigma occurs when a label associated with a negative stereotype is attributed to an individual characteristic, causing people with such characteristics to be seen as separate and of lower status compared with people without the characteristic.[4] When stigma occurs, 'power' is exercised by stigmatisers to keep stigmatised groups 'down' or dominated/exploited; 'within', to maintain social norms; and 'away' by means of social exclusion.[5] In dementia, stigmatisation occurs through negative stereotypes, related to cognitive decline can lead to depersonalisation and considering the person as unable to continue to live in and contribute to society.[6–9] Although not every person living with dementia is an

## BACKGROUND

There are approximately 55.2 million people living with dementia worldwide and nearly 70% of this population live in low-income and middle-income countries (LMICs), such as Brazil.[1] By 2050, this number is expected to increase to 139 million.[1] Stigma and discrimination related to dementia are common, which is particularly important among healthcare workers, considering the detrimental

older person, older people living with dementia are likely to experience stigma related to dementia as well as from ageism and ableism, further impacting their rights and well-being.[10]

Several studies provide robust evidence on the negative impact that stigma and discrimination has on people living with mental disorders more broadly.[11–13] Although there is a paucity of research on the impact of stigma on people living with dementia,[6] the existing evidence shows that stigma can lead to negative feelings about one self, shame, symptom and diagnosis concealment, negative social interactions, reduced access to care networks and social participation and even suicide.[2 6 9 14–16] A global survey on attitudes towards dementia among members of the public, people living with dementia, family carers and healthcare professionals from 155 countries showed that over 85% of respondents living with dementia had their opinion not taken seriously as well as between 35% and 57% (in high-income countries and LMICs, respectively) had been treated unfairly in intimate relationships. Moreover, 40% of the general public believed health professionals ignore people living with dementia; and 35% of carers reported hiding the diagnosis of dementia of a family member.[2]

Limited knowledge about dementia as well as widespread misbeliefs (eg, believing that dementia is a natural part of ageing) and negative attitudes (eg, considering people living with dementia are burdensome) may make health professionals less likely to detect dementia and provide adequate care to people affected by this condition.[3 9] In over 75% of cases globally, diagnosis of dementia is either not made, or is made at a later stage, when the person living with the condition is no longer able to make decisions about their life and well-being independently.[3] Structural forms of stigma—such as when systems, policies or services are designed in a way that discriminates directly or indirectly against people living with dementia[17]—also contributes to poor healthcare provision in dementia, by means of lack of investment in appropriate services and in training for health and social care providers, who continue to be unprepared to diagnose and to provide care and support for people living with dementia,[3] further contributing to the experiences of stigma and discrimination experienced by these individuals. The current state of healthcare systems and providers can, therefore, be seen as a consequence as well as a source of stigma related to dementia.

A recent systematic review with 56 studies showed that most interventions to reduce mental health-related stigma in LMICs are effective.[18] These were focused on stigma related to schizophrenia, suicide, depression, child and adolescent mental health, bipolar disorder, anorexia nervosa and post-traumatic stress disorder among healthcare workers, students or on more than one group. Around 75% found a significant positive effect for all main stigma outcomes and 25% found a small positive effect for some but not all stigma outcomes. Among the moderate or high-quality studies (n=38), 10 found a significant long-term positive effect on stigma outcomes.

The most common approach is education; however, a wide variety of methods—from creative arts-based approaches to those which emphasise empowering people with mental health conditions—can be successful, including social contact interventions.[18] Currently, very little is known about interventions which reduce stigma related to dementia[19] and research on knowledge and attitudes related to dementia in Latin American countries is scarce.[20 21] There is an urgent need for further research on dementia-related stigma in different contexts and cultures.[1 6]

It has been estimated that 77% of people living with dementia in Brazil are not diagnosed.[22] Prevalence estimates of people living with dementia vary from 5.1% to 19.0% among those aged 65 and over.[23] Brazil still lacks well-funded and well-equipped health systems to meet the needs of the growing population of people living with dementia and their families. The country has a universal health system in which primary care units (PCUs) are the first point of access for people who experience any physical or mental health problem. In most regions, PCUs have community health workers (CHWs) who contribute to health promotion and disease prevention activities.[24] Most CHWs live in the same communities where they work, being a valuable point of contact and an intersection between health services and their communities. Health systems with CHWs have the potential to help expand the delivery of mental healthcare and close the existing mental health gap in LMICs.[25] However, the training activities offered to CHWs in Brazil is focused on the control of communicable diseases, maternal and child health and non-communicable physical diseases, such as diabetes and hypertension, with very little training dedicated to dementia. Providing CHWs with adequate knowledge on dementia, as well as on common stigmatising and discriminatory practices, and positive attitudes and behaviours towards people living with dementia may help increase the number of people living with dementia identified by the PCU as well as the quality of the healthcare and support provided to these individuals.

## AIM AND OBJECTIVES

The overarching aim is to design an antistigma intervention to reduce stigma and discrimination towards people living with dementia among CHWs in the city of São José dos Campos, Sao Paulo, Brazil and to pilot the intervention using a feasibility randomised controlled trial. The specific objectives are:

1. To develop a group-based intervention to improve knowledge about dementia as well as attitudes and behaviours towards people living with dementia.
2. To test the intervention in a randomised controlled feasibility study and evaluate the acceptability and feasibility of performing such an intervention in a future randomised controlled trial.

## METHODS AND ANALYSIS

### Study design

A randomised controlled feasibility study will be conducted as part of a large multinational research programme (*Strenghtening Responses to Dementia in Developing Countries: STRiDE - www.stride-dementia.org*), which aims to contribute to the improvement of care systems, treatment and support for people living with dementia and their families in Brazil and other LMICs. An exploratory qualitative study was conducted previously involving semistructured interviews and focus groups with people living with dementia, family carers, healthcare workers and members of the public in three large cities in the state of Sao Paulo.[9 26] The findings from the study helped inform the content and design of this intervention (table 1 and figure 1). This protocol (V.1; date: 4 May 2022) was prepared in line with the Standard Protocol Items: Recommendations for Interventional Trials checklist (online supplemental material I).[27 28] The study protocol will be registered in the Brazilian National Registry of Clinical Trials (REBEC: https://ensaio-sclinicos.gov.br/ or in ClinicalTrials.Gov https://clinicaltrials.gov/), which will include all items from the WHO Trial Registration Data Set (https://www.who.int/clinical-trials-registry-platform/network/who-data-set).

### Participants and sampling

All CHWs in each of the 20 eligible PCUs located in São José dos Campos will be invited to take part. Individuals working as a CHWs for less than 3 months will be excluded. Each PCU has approximately from 8 to 12 CHWs, totalling 160 to 240 potential participants. A proportion (30%) of refusals to participate and dropouts are expected (eg, change in work schedules); therefore, we estimate to include approximately 14 PCUs involving around 150 participants in the trial.

### Method of generating the allocation sequence

Randomisation and allocation will be conducted by blinded member of the research team at the PCU level using computer-generated random numbers. The PCUs whose managers agree to participate will be randomly allocated in two parallel groups—experimental group or control group—so that all CHWs of a given PCU who voluntarily accept to take part can be allocated in the same group forming a cluster and reducing the potential for contamination (eg, sharing knowledge learnt in the intervention between PCUs). The allocation will be made randomly at 1:1, so that there is the same number of PCUs in each group, offering as much as possible balanced participation among CHWs from PCUs located in the various geographical regions, proportional to the total number of PCUs of each region (20%: north, 20%: south, 20%: east, 20%: west, 20%: centre). The characteristics of participants (eg, age, gender, time working as CHW) will be compared between groups and controlled statistically if significant differences are observed.

Before informing the PCUs about the group to which their CHWs have been allocated, every CHWs wishing to participate will read and sign a consent form. Recruitment and enrolment will be conducted by a separate member of the team, who will be blinded to the allocation. Then, all participants will be invited to complete the baseline measures (t1). The experimental group will be informed about the dates and timings of the intervention and the control group will be told that they will be allocated to a waiting list. After the completion of the final assessments from both experimental group and control group (t2), the intervention will be offered to the control group.

### Development of the content of the intervention

Participants within the experimental group will receive a group intervention created by the researchers—DO (Nurse, Brazil), CG (Gerontologist, Brazil), CF (Psychiatrist and Epidemiologist, Brazil), FAFdM (Physiotherapist, Brazil), EM (Linguist, Brazil), ACAF (Journalist, Brazil), NF (Psychologist and dementia researcher, England) and SE-L (Stigma expert, England). The researchers have extensive experience in research and clinical practice with people living with dementia, CHWs and family carers. The intervention materials were informed by a rapid scoping review conducted by the team on dementia-related stigma in Latin America and the Caribbean (unpublished) and on the exploratory qualitative work conducted previously with people living with dementia, family carers, healthcare providers (including CHWs) and members of the public in Brazil.[9 26] This was also supplemented with the literature on ethical and patient-centred care for people living with some type of mental and/or neuropsychiatric disorder,[29] literature in the area of person-centred care for people living with dementia[30–33] as well as by evidence for effective interventions to reduce stigma in LMICs.[11 13 18]

The researchers will use audio-visual and printed materials containing, for example, videos of people with dementia and carers sharing their personal experiences, reflexive activities, group discussions and presentations through Power Point (table 1).

The group intervention will seek to improve knowledge about dementia as well as attitudes and behaviours towards people living with dementia and their carers. Figure 1 presents the theory of change to be tested in this intervention, including 'causes of problems', 'problems', 'resources', 'activities or actions', 'mechanisms of change' and 'expected outcomes'. This logic model was developed based on the literature related to stigma and dementia as well as on exploratory work conducted previously in Brazil including experts by experience.[9 26] This has been presented and discussed among the international and multidisciplinary team (*Strenghtening Responses to Dementia in Developing Countries: STRiDE - www.stride-dementia.org*) for internal validation.[34]

### Active intervention and control

The intervention will be undertaken in different sessions with the participants of each PCU allocated to the

**Table 1** Summary of the intervention components and activities

| When | | Why: what we want to achieve | What: topic covered | How: method or strategy |
|---|---|---|---|---|
| **Day 1:** Building knowledge and demystifying dementia: beginning the transformation process. | Session 1 | ► To understand the general structure of the intervention and get to know one another, which will be important to build a non-judgmental space of trust among the individuals and for their spontaneous contribution to the activities.<br>► To get in touch with individual and shared beliefs and questions related to dementia, and to gain a better understanding of the condition and of the 'individual behind the disease'. | ► What is dementia?<br>► What is not dementia?<br>► How people living with dementia are seen? | ► Before the session starts, we ask participants to individually write down their views on dementia and people living with dementia using directed questions.<br>► We provide an interactive presentation on knowledge and common beliefs related to dementia.<br>► Each participant reads their views and shares with the group voluntarily. The group reflects and discusses these based on the presentation delivered earlier. |
| | Session 2 | ► To understand what dementia is, what dementia is not, and what treatment and care possibilities exist.<br>► To understand how dementia can affect people living with dementia and their families.<br>► To experience, through group dynamics, the impacts caused by prejudice, discrimination, and negative language on the lives of people living with dementia. | ► What is dementia?<br>► What is not dementia?<br>► Known and unknown, modifiable, and non-modifiable risk factors.<br>► Pharmacological and non-pharmacological interventions can help people who live with dementia and their carers to have quality of life.<br>► Management and control of dementia symptoms. | ► This will be a dynamic session in which we ask questions to the group in one slide, and provide feedback in the following slide, and so on.<br>► After, we will have the "secret box" dynamic: we will distribute individual boxes containing two photos of famous people and a mirror at the end. We ask participants to describe the characteristics of the two people and then to describe themselves. We try to identify differences in how they would describe 'others' and how they would describe 'themselves' to stimulate empathetic and non-judgemental approaches.<br>► We will hold a reflective and introductory debate on stigma, prejudice, discrimination, and use of language.<br>► We will promote reflection on how the participants felt when placing themselves in the 'shoes' of someone living with dementia. |
| **Day 2:** Breaking down labels and stereotypes and improving attitudes towards people living with dementia. | Session 1 | ► To understand how thoughts, feelings, attitudes, and behaviours of other people affect people living with dementia.<br>► To understand that the consequences of stigma and discrimination that people living with dementia experience are as important as knowing the disease itself.<br>► To understand the power of language as a mechanism of prejudice and discrimination.<br>► To identify possible behaviours and inadequate attitudes towards dementia and people living with dementia among participants. | ► The importance of language, what is said and what is not said, in relation to people living with dementia, as well as common stigma and discrimination practices.<br>► Commonly used terms that reproduce stigma. | ► This will be a dynamic session in which we ask questions to the group in one slide, and provide feedback in the following slide, and so on.<br>► Re-read and hold a debate on the answers participants gave on the first day, prior to the intervention start and compare them with the views they have now about people living with dementia and their carers. |
| | Session 2 | ► To understand that people living with dementia have desires, preferences, feelings, and aspirations.<br>► To sensitise participants about common negative behaviours, thoughts, and attitudes towards people living with dementia.<br>► To develop empathy for the everyday issues experienced by people living with dementia. | ► The heterogeneity of people living with dementia.<br>► Importance of looking at people who live with dementia in a holistic way, beyond the disease.<br>► Compassion, understanding, and empathy for everyday situations experienced by people living with dementia. | ► Show video narratives of people living with dementia<br>► Hand out two case vignettes to participants in which we depict two cases of people who develop dementia, and they have to read these in 'first person' as if the story described were theirs<br>► Hold a group discussion about how they felt in reading those experiences and about how they would have liked to be treated. |

Continued

**Table 1** Continued

| When | | Why: what we want to achieve | What: topic covered | How: method or strategy |
|---|---|---|---|---|
| **Day 3:** Developing compassion and empathy and strengthening communication and behavioural skills. | Session 1 | ► To explain and demonstrate positive verbal and non-verbal communication strategies.<br>► To apply these strategies in groups.<br>► To actively try to recognise possible transformations in their beliefs, attitudes, and behaviours related to people living with dementia compared with the beginning of the intervention. | ► Practical strategies for reflection on appropriate ways of caring for people living with dementia.<br>► Positive verbal and non-verbal communication strategies. | ► Content display using slides. |
| | Session 2 | ► To identify inappropriate behaviours and attitudes in their own practices.<br>► To reflect on how to apply positive verbal and non-verbal communication strategies learnt in the previous session.<br>► To generate feedback on the six sessions held. | ► Importance of the CHWs work in combating the stigma of dementia. | ► Real-life-based stories about the daily life of people living with dementia are presented and discussed.<br>► Reflections and discussion of the theoretical contents discussed in the 3 days of intervention. |

CHWs, community health workers.

experimental group. The intervention will be led by DO and will consist of three group meetings, held on 3 consecutive days, lasting 3 hours each, involving all participating CHWs from each PCU in each meeting (total=9 hours over 3 days for all CHWs) (figure 2). This schedule was organised in a way that would improve participant's adherence to intervention as they will need to be off duty during the study sessions. The control group will not receive any activity and will continue with their usual routine. We will deliver the same activity after the end of the study to individuals from the control group who wish to receive it.

## Assessment
### Primary and secondary outcomes
The primary outcome will be the feasibility of the intervention, including the findings of a nested qualitative component. The secondary outcomes will be the stigma-related measures: knowledge, attitudes and intended behaviour. All the outcome measures are described as follows.

### Baseline and poststudy assessment
Baseline (t0) and poststudy assessments (t2) will be the same for both experimental and control groups.

| General parameters | Type of program<br>Feasibility study | | Modality<br>Psychoeducational | | Target population<br>Community health workers | Mode<br>Group based | Local<br>Primary healthcare units | City<br>São José dos Campos, São Paulo |
|---|---|---|---|---|---|---|---|---|
| **Problems** | **Poor knowledge and beliefs**<br>○ Dementia is a natural part of aging<br>○ Dementia only affects older people<br>○ Every person who lives with dementia was born with a predisposition to dementia<br>○ The person is to blame for the development of dementia (e.g., lifestyle)<br>○ The brain has not developed<br>○ Nothing can be done to support people living with dementia and their carers | **Stereotypes and labelling**<br>Views that people living with dementia are:<br>○ Dependent and incapable<br>○ Passive<br>○ Depressed and fragile<br>○ Isolated and alone<br>○ Crazy and unpredictable<br>○ Aggressive<br>○ Difficult to deal with<br>○ Burdensome<br>○ Repetitive<br>○ Abnormal<br>○ Plaintiffs<br>○ Manipulative<br>○ Unable to have feelings | | **Separation, blame, avoidance, loss of status, social exclusion**<br>○ The person is not allowed to make decisions and to engage in social\alone activities and is discouraged from maintaining social bonds and roles (e.g., work)<br>○ The person is no longer considered to be part of groups to which they were before dementia<br>○ The person has the value of their opinion and contribution reduced, and is ignored or avoided<br>○ The person is expected to obey and 'behave' and has their freedom removed and\or restricted<br>○ The person is blamed for the stigma and discrimination (e.g., excluded due to being considered repetitive)<br>○ The person suffers physical and emotional abuse<br>○ There is a reduction in social ties and participation<br>○ Carers feel isolated and excluded<br>○ Carers tend to keep the person living with dementia indoors to "protect" them | | **Internalized stigma**<br>○ Low self-esteem<br>○ Doubt own feelings and experiences<br>○ Denial of dementia<br>○ Diagnosis concealment<br>○ Social withdrawn<br>○ Loss of identity<br>○ Reduced wellbeing | **Less access to health services**<br>**Poor quality services**<br>○ Lack of support services<br>○ Lack of training and qualified professionals<br>○ Reduced help seeking<br>○ Late diagnosis<br>○ Care mostly focused on physical needs<br>○ The care provided generates dependence and is passive<br>○ Care does not involve the visions and desires of the person living with dementia and their carers<br>○ Carers have high emotional and physical burden |
| **Intervention resources** | Audio-visual presentations | | Video-narratives of people living with dementia and caregivers | | Group discussions and reflections | | Dosage: Six sessions, divided into three sequential meetings, lasting three hours per day (9h in all) | |
| **Activities** | What is dementia and how does it affect people living with dementia and their carers? | Experiences of people living with dementia and their carers. | Stigma, stereotypes, attributes, and language: discriminatory practices and their consequences. | Human rights-based practices, person-centred, and ethical care. | Strengthening tangible skills of compassion, empathy, and communication (verbal and nonverbal). | | How can I improve my CHWs practice to better serve people living with dementia and their carers?<br>Brief and general evaluation of the programme | |
| **Outcomes 30 days after intervention** | Improvement of knowledge and attitudes | | Improved attitudes towards people living with dementia and their caregivers | | Development of person-centred skills | | Reduction of stigma levels | Reduction of discrimination levels |

**Figure 1** Theory of change: stigma and discrimination related to dementia among community health workers.

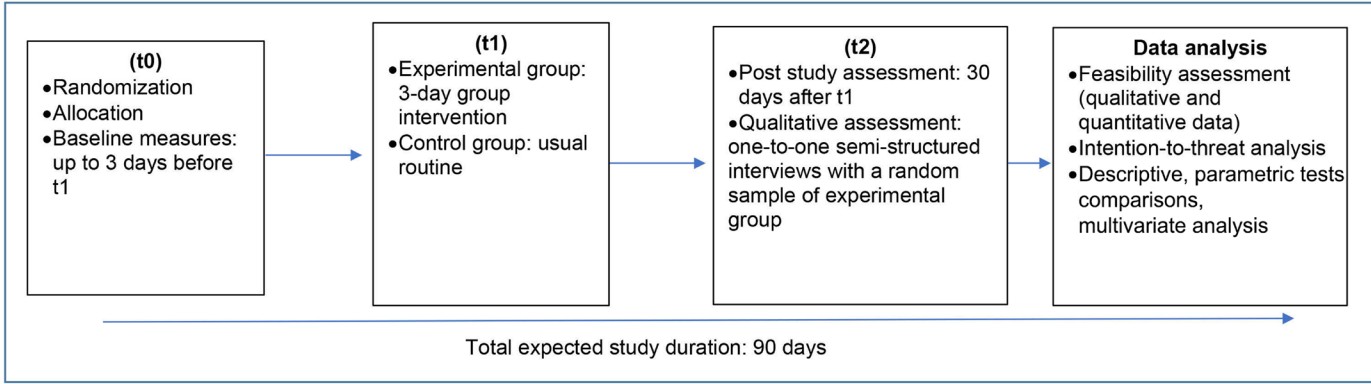

**Figure 2** Study flow.

Additional feasibility questions (closed and open questions) will be applied to the experimental group on the last day of the intervention as well as in t2, and 15–20 participants in the experimental group will be invited to participate in one-to-one semistructured interviews. Apart from the qualitative interviews, the written questions are all self-administered and will be completed on paper. Questionnaires will be distributed and collected by an assistant researcher who will be blinded to participant allocation. We hope that using anonymous self-administered questionnaires (identifiable only through participant code) will help mitigate any potential risk for social desirability bias. We will also highlight to participants that the questionnaires are not identifiable and that any answers they provide will not be linked to their names, and that they should be free to answer honestly to all the questions without any fear of being judged. The researchers applying the intervention will not have access to the participants' responses until the end of data collection. It is expected that CHWs will take 10 min to 15 min to complete all the questions. Considering the acceptance of approximately 14 PCUs to participate in the study, it is estimated that the total period between t0 and t2 in all PCUs will be 90 days.

### Sociodemographic and stigma-related outcomes

Sociodemographic questions will include age, gender, level of education, religion as well as information on previous training and experience with dementia (details in online supplemental material II). Stigma-related questions (dementia knowledge, attitudes and intended behaviour towards people living with dementia) were derived from the 2019 Global Stigma and Dementia Survey (WAR). The WAR questionnaire has been administered to more than 70 000 people worldwide, including healthcare professionals, members of the public, people living with dementia and their carers, through which process it has been translated to Brazilian Portuguese.[2] The WAR questions were designed in Likert scale format and are concerned with the dementia knowledge, attitudes and anticipated behaviour towards people living with dementia. The questions were informed by existing validated stigma and discrimination scales[35 36] and have

been validated into a reduced number of items with high psychometric performance.[2 37]

### Feasibility assessment

First, a brief evaluation will be conducted on the last day of the intervention through discussion and notes will be taken by the researchers (eg, relevance, positive and negative aspects). Then, in t2, additional measures (objective and subjective) regarding the feasibility of the intervention will be applied, including a measure of satisfaction with the intervention[38] and open questions (details in online supplemental material II). We will also collate information on recruitment rate and retention overall and per session, intervention completion rate, evaluation measures completion, appropriateness/acceptability and fidelity of the intervention. Provisional decision rules for what procedures to carry through to the full trial will include: retention of at least 70% completing at least two-thirds of the sessions, less than 15% missing on outcome measures and fidelity of at least 75% according to a fidelity checklist to be completed by a member of the research team in every session.[39–41] Qualitative and quantitative findings will be collated and reported in a transparent, and the decision regarding whether the intervention is feasible or not will be taken (plus justified and reported) on discussion among the research team.

### Nested qualitative component

Two participants from each PCU participating in the experimental group (n~14), including those who dropped out the study, will be invited to voluntarily participate in individual semistructured interviews to explore further aspects of feasibility and potential impact of the intervention (details in online supplemental material III). Interviews will take place on the same day of t2, after all the poststudy measures have been completed. The sample will be selected purposively, aiming to include a variety of sociodemographic characteristics, locations and background training on dementia. To reduce the potential selection bias based on the interpersonal experience of the researchers applying the intervention with the participants, another member of the team will make this selection based on the characteristics reported in the

questionnaires, and another researcher will conduct the interview itself. The interviews will last a maximum of 60 min, will be voice recorded, and will be held in a private room at the PCUs where the CHWs work.

## Data analysis

We will seek to adhere to the steps proposed in the modelling phase of the Medical Research Council guidelines for developing complex interventions.[42] The purpose of a feasibility study is not to make a formal analysis of the primary outcome, but to evaluate trial processes to determine whether to progress to a study of effectiveness and to estimate parameters needed to design the future trial.[42] Data analysis will be conducted by a researcher who has not been involved with the intervention itself and who is blinded to the arm allocation. Double data entry and checking will be used to ensure accuracy.

We hypothesise that the intervention is feasible and might be able to improve knowledge, attitudes and intended behaviour in relation to dementia among CHWs. Analysis of quantitative data will be based on 'intention to treat', that is, data from all participants will be included regardless of their withdrawal from the study or not. Descriptive analysis will include central and dispersion measures, according to the types (eg, continuous, categorical, nominal) and distribution patterns of the variables, including number of incomplete responses and dropouts, ceiling and floor effects. Differences in levels of variables related to dementia stigma, namely, knowledge, attitudes and intended behaviours towards people living with dementia, before and after the intervention, between groups and between time points (t0 and t2), will be analysed using Student's t test or a non-parametric test, according to data distribution. We will measure internal reliability (Cronbach's alpha) and test–retest, levels of variance, correlation between sociodemographic variables and between the different outcome variables. An appropriate statistical approach will be chosen to handle any missing data depending on its pattern and type of variables.

Quantitative feasibility measures will be analysed descriptively and will be compared with the results from the outcome measures. Feasibility data collected through qualitative methods will be analysed in an integrated manner using content analysis as well as triangulation techniques in NVivo.[43 44] In preparation to that, semistructured interviews will first be transcribed anonymously and verbatim.

## Ethics and dissemination

This study was approved by the National Commission for Ethics in Research in Brazil (CONEP) (*n. 5.510.113*). Every individual taking part will be informed about their rights as participants, including the fact that non-participation will not affect in any way their work status or care received at the PCU. Every participant will sign a consent form; those participating in the semistructured interviews will be required to sign a second consent form specific for this research activity.

Safety monitoring procedures have been created to protect participants and researchers, including safety measures to prevent emotional impact and the spread of COVID-19 (eg, mandatory vaccination and use of mask during any study activity as well as physical distance during the study. All personal information about potential and enrolled participants will be collected, shared and maintained in line with the norms and regulations of the Brazilian National Ethics Committee (http://www.conselho.saude.gov.br/comissoes-cns/conep) regarding confidentiality measures stated in order to protect confidentiality before, during and after the study. Any important protocol modifications will be immediately communicated to the Research Ethics Committee and the study will be stopped until an approval is obtained for the study to continue. The study will not have a data monitoring committee as this is not required for a feasibility non-pharmacological study in Brazil.

We plan to disseminate this study in open-access scientific and community events related to dementia. The intervention materials will be published online and will be available for use by anyone who wishes to translate, adapt and implement it. Our research team works closely with policymakers as part of a national advisory team, which we hope will help increase the possibility of such intervention to be applied in more settings in Brazil. An antistigma intervention toolkit will be informed by this study will be accessible nationally in Brazil and globally through open-access publication to support researchers and practitioners with the implementation of dementia-related antistigma actions.

## Patient and public involvement

Through a series of focus groups and semistructured interviews, we have explored the views of family carers, people living with dementia, healthcare workers and members of the public about stigma and discrimination related to people living with dementia. The findings from this exploratory activity have informed the development of the intervention. We have also presented and discussed this intervention protocol among the international and multidisciplinary team (*Strenghtening Responses to Dementia in Developing Countries: STRiDE - www.stride-dementia.org*) for internal validation.[34] As this is a protocol for a feasibility study, which involves both quantitative and qualitative procedures, we aim to gather participants' views about the intervention as part of the study itself in order to improve it for a future trial. We will also continue to consult our international consortium, which includes experts by experience, about the various methodological and practical aspects of the work as it progresses.

## Planned study dates

We plan to start the study in August 2022 and to end it in January 2023.

## DISCUSSION

The number of people living with dementia in Brazil is likely to increase in the next decades; however, the number of people who are undiagnosed remains high and there is limited access and availability of appropriate support. Improving knowledge, attitudes and behaviour of healthcare workers towards people living with dementia is a global health priority. In doing so, this is likely to reduce the stigma and discrimination experienced by people living with the condition, protect the quality of their lives, improve diagnosis rates and quality care. Antistigma interventions have been successful in other countries and disease contexts, such as other mental health conditions and HIV. However, there is a paucity of research on antistigma interventions related to dementia, particularly targeting healthcare workers.

To our knowledge, this is the first antistigma intervention related to dementia to be conducted in Brazil and we are not aware of any other similar intervention in other Latin American countries. We will target CHWs who are the first point of healthcare access for people living with dementia in the community and we hope that this will help increase the number of people living with dementia who are identified and attended by the healthcare systems. As CHWs work alongside multidisciplinary teams, we hope that the learning experiences achieved through this intervention will be shared among other members of the team, contributing to a further reach of the intervention and higher impact. The measures of feasibility included in this protocol will be paramount to ensure that the invention is acceptable, relevant and effective to be applied in a future randomised controlled trial in Brazil and in other LMICs. An antistigma toolkit will be created from the intervention will also help ensure fidelity of future trials as well as applicability in PCUs whose teams have the opportunity to apply it in their territories.

**Author affiliations**
[1]Department of Psychiatry, School of Medicine, Universidade Federal de São Paulo, Sao Paulo, Brazil
[2]Brazilian Federation of Alzheimer's Associations (FEBRAZ), Paraná, Brazil
[3]Applied Linguistics, Department of Modern Languages, Universidade Estadual de Londrina (UEL), Paraná, Brazil
[4]Centre for Dementia Studies, Brighton and Sussex Medical School, Brighton, UK
[5]Personal Social Services Research Unit, London School of Economics and Political Science, London, UK
[6]Health Technology Assessment Unit, Hospital Alemão Oswaldo Cruz, Sao Paulo, Brazil

**Contributors** DO has led the design and development of the intervention, and the writing of the manuscript; CG has supported with the development of the intervention and has contributed to the writing of the manuscript; FAFdM, EM, ACAF, NF, SE-L and CF have supported with the design and development of the intervention and have contributed to the writing of the manuscript.

**Funding** This project is sponsored by UK Research and Innovation Global Challenges Research Fund (ES/P010938/1) (https://www.ukri.org/what-we-offer/collaborating-internationally/global-challenges-research-fund/). The study sponsor had no role in study design; collection, management, analysis, and interpretation of data; writing of the protocol; and the decision to submit the protocol for publication, nor they had ultimate authority over any of these activities.

**Competing interests** None declared.

**Patient and public involvement** Patients and/or the public were involved in the design, or conduct, or reporting, or dissemination plans of this research. Refer to the Methods section for further details.

**Patient consent for publication** Not applicable.

**Provenance and peer review** Not commissioned; externally peer reviewed.

**Data availability statement** This is a protocol paper and therefore no data have been collected yet. For more information about the study, please contact Dr Deborah Oliveira (oliveiradc.phd@gmail.com) or Dr Sara Evans-Lacko (s.evans-lacko@lse.ac.uk).

**ORCID iDs**
Déborah Oliveira http://orcid.org/0000-0002-6616-533X
Nicolas Farina http://orcid.org/0000-0002-0635-2547

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
