## [Reviewer comments · BMJ Open]

ARTICLE DETAILS

TITLE (PROVISIONAL)	Reducing dementia-related stigma and discrimination amongst community health workers in Brazil: Protocol for a randomized controlled feasibility trial
AUTHORS	Oliveira, Deborah; Godoy, Carolina; da Mata, Fabiana; Mateus, Elaine; Franzon, Ana Carolina; Farina, Nicolas; Evans-Lacko, Sara; Ferri, Cleusa

VERSION 1 – REVIEW

REVIEWER	Albert, Steven M. University of Pittsburgh, Behavioral and Community Health Sciences
REVIEW RETURNED	01-Jan-2022

GENERAL COMMENTS	1 .The focus on feasibility is reasonable, but the proposed measures and their treatment could be described in greater detail. For example, the authors mention "number of sessions carried out according to protocol, total number of participants and per session, participant characteristics, recruitment rate, intervention completion rate, evaluation measures completion, appropriateness/acceptability, and fidelity of the intervention." What criteria will be used to determine feasibility? Greater than 80% attendance, for example? How exactly will fidelity, a delivery indicator, be assessed? 2. No analysis is proposed for comparing the study arms on the sigma measures. 3. The measures will likely elicit strong social desirability biases. Can the authors oppose ways to handle this bias?
---

REVIEWER	Arai, Asuna Hokkaido University
REVIEW RETURNED	22-Feb-2022

GENERAL COMMENTS	This research would be expected to provide useful information on feasibility of interventions to CHWs to reduce stigmatization towards people living with dementia. There are some comments below. Abstract Methods and analysis, Please give the time of follow-up assessments. Background
--

	P4, L3-5, The authors mentioned about a recent systematic review. To whom are most interventions effective to reduce mental health related stigma in LMICs? Please clarify it. P4, L34-38, As the authors explained, the reviewer understands that CHWs in Brazil have very little training dedicated to dementia. However, how much stigma and discrimination towards people living with dementia do CHWs actually have? Are there any previous reports? Methods and analysis P5, L55-, The authors describe, "It will also be taken into account, as far as possible, balance between the characteristics of the participants (e.g. age, gender, time working as CHW)." How did authors balance between the characteristics? Data analysis P13, L17-22, " Differences in levels of dementia knowledge, as well as attitudes and intended behaviours towards people living with dementia before and after the intervention between groups and between time points (t0 and t2) will be measured using Student's t test or a non-parametric test..." This analysis is important to evaluate the feasibility of the intervention in terms of its effect to improve knowledge, attitudes, and behaviours. The reviewer wonders if any confounding factors would be considered and recommends to use multivariable analysis including possible confounders. Figure 2. Study flow is too simple. More detailed information should be included in the flow chart. That would be helpful for readers.
--	---

VERSION 1 – AUTHOR RESPONSE

Reviewer: 1. Dr. Steven M. Albert, University of Pittsburgh	
1. The focus on feasibility is reasonable, but the proposed measures and their treatment could be described in greater detail. For example, the authors mention "number of sessions carried out according to protocol, total number of participants and per session, participant characteristics, recruitment rate, intervention completion rate, evaluation measures completion, appropriateness/acceptability, and fidelity of the intervention." What criteria will be used to determine feasibility? Greater than 80% attendance, for example?	Thank you for your suggestions. We have clarified this on page 13. "We will also collate information on recruitment rate and retention overall and per session, intervention completion rate, evaluation measures completion, appropriateness/acceptability, and fidelity of the intervention. Provisional decision rules for what procedures to carry through to the full trial will include: retention of at least 70% completing at least two-thirds of the sessions, less than 15% missing on outcome measures and fidelity of at least 75% according to a fidelity checklist to be completed by a member of the research team in every session³⁹⁻⁴¹ Qualitative and quantitative findings will be collated and reported in a transparent, and the decision regarding whether the intervention is feasible or not will be taken (plus justified and reported) upon discussion among the research team."

How exactly will fidelity, a delivery indicator, be assessed?	
2. No analysis is proposed for comparing the study arms on the sigma measures.	This had been included on page 14. We revised this part to increase clarity: “Differences in levels of variables related to dementia stigma, namely knowledge, attitudes and intended behaviours towards people living with dementia, before and after the intervention, between groups, and between time points (t0 and t2), will be analysed using Student's t test or a non-parametric test, according to data distribution.”
3. The measures will likely elicit strong social desirability biases. Can the authors oppose ways to handle this bias?	Thank you for pointing this out. We had included on page 12 that “apart from the qualitative interviews, the written questions are all self-administered and will be completed on paper.” We hope that the use of self-administered questionnaires will help mitigate any potential risk for social desirability bias. We have modified the text to highlight this (page 12): “Apart from the qualitative interviews, the written questions are all self-administered and will be completed on paper. Questionnaires will be distributed and collected by an assistant researcher who will be blinded to participant allocation. We hope that using anonymous self-administered questionnaires (identifiable only through participant code) will help mitigate any potential risk for social desirability bias. We will also highlight to participants that the questionnaires are not identifiable and that any answers they provide will not be linked to their names, and that they should be free to answer honestly to all the questions without any fear of being judged.”
Reviewer: 2. Dr. Asuna Arai, Hokkaido University	
Abstract: Methods and analysis: please give the time of follow-up assessments.	Thank you. We have included this information in the abstract.
Background: P4, L3-5, The authors mentioned a recent systematic review. To whom are most interventions effective to reduce mental health related stigma in LMICs? Please clarify it.	We have added more information about this review on page 4, as recommended. “A recent systematic review with 56 studies showed that most interventions to reduce mental health related stigma in LMICs are effective.⁴ These were focused on stigma related to schizophrenia, suicide, depression, child and adolescent mental health, bipolar disorder, anorexia nervosa and post-traumatic stress disorder among health care workers, students, or on more than one group. Around 75% found a significant positive effect for all main stigma outcomes and 25% found a small positive effect for some but not all stigma outcomes. Among the moderate or high-quality studies (n = 38), ten found a significant long-term positive effect on stigma outcomes.” Unfortunately, no previous study has been conducted to measure stigma and discrimination related to dementia among CHW in Brazil. Prior to preparing this protocol, we conducted an

P4, L34-38, As the authors explained, the reviewer understands that CHWs in Brazil have very little training dedicated to dementia. However, how much stigma and discrimination towards people living with dementia do CHWs actually have? Are there any previous reports?	exploratory qualitative study (mentioned on pages 5, 6, 10, 15), of which part has been published (carers and people living with dementia, and part has not been published yet (community health workers and members of the public). We found several elements of stigma which have helped inform the intervention protocol (Tables 1 and 2):  • in the (unpublished) data from CHW and members of the public, we found that dementia was understood as an unknown 'state' of malfunctioning, and that people living with dementia were seen as unhuman, burdensome, and unworthy. We also found elements of undefined care and abusive behaviours towards people living with dementia; and • in the (published) study with carers and people living with dementia, experiences of stigma and discrimination were reported in a variety of domains. And of course, we also considered the literature of stigma related to dementia among healthcare workers in other low- and middle-income countries within our international consortium (such as: https://journals.sagepub.com/doi/10.1177/14713012211014800), though cultural differences are likely to play a role in how stigma is manifested.
Methods and analysis: P5, L55-, The authors describe, "It will also be taken into account, as far as possible, balance between the characteristics of the participants (e.g., age, gender, time working as CHW)." How did authors balance between the characteristics?	We have clarified this on page 6: "The characteristics of participants (e.g., age, gender, time working as CHW) will be compared between groups and controlled for statistically if significant differences are observed."
Data analysis: P13, L17-22, " Differences in levels of dementia knowledge, as well as attitudes and intended behaviours towards people living with dementia before and after the intervention between groups and between time points (t0 and t2) will be measured using Student's t test or a non-parametric test..." This analysis is important to evaluate the feasibility of the intervention in terms of its effect to improve knowledge, attitudes, and behaviours. The reviewer wonders if any confounding factors would be considered and recommends using multivariable	Although we believe that using a randomized sample should suffice to reduce any risk of confounders, we have included this to our analysis plan on page 6: "We will also use multivariate analysis to detect any possible confounders to the analysis, such as age, sex, religion, time working as a CHW, close personal contact with a person living with dementia and having previous training in dementia."

analysis including possible confounders.	
Figure 2. Study flow is too simple. More detailed information should be included in the flow chart. That would be helpful for readers.	Thank you we have added some more information to the figure, as suggested.